# Flame Retardant Nano-Structured Fillers from Huntite/Hydromagnesite Minerals

**DOI:** 10.3390/nano12142433

**Published:** 2022-07-15

**Authors:** Konstantinos S. Andrikopoulos, Giannis Bounos, Georgia Ch. Lainioti, Theophilos Ioannides, Joannis K. Kallitsis, George A. Voyiatzis

**Affiliations:** 1Foundation for Research and Technology-Hellas, Institute of Chemical Engineering Sciences (FORTH/ICE-HT), GR-26504 Patras, Greece; ioannis.bounos@gmail.com (G.B.); glainioti@upatras.gr (G.C.L.); kallitsi@upatras.gr (J.K.K.); gvog@iceht.forth.gr (G.A.V.); 2Department of Physics, University of Patras, GR-26504 Patras, Greece; 3Department of Chemistry, University of Patras, GR-26504 Patras, Greece

**Keywords:** nano-sized Mg(OH)_2_, hydromagnesite/huntite minerals, calcination/hydrothermal synthesis, flame retardant materials, optical properties

## Abstract

In the current study, we propose a simple hydrothermal pathway to synthesize nano-structured Mg(OH)_2_ after application of thermal decomposition followed by hydration of commercial minerals based on hydromagnesite and huntite. The synthesis of nano-materials is performed without the use of any catalyst. The effect of decomposition temperature on the hydrothermal synthesis of Mg(OH)_2_ is extensively studied. It is shown that the morphology of resulting structures consists typically of particles ~200 nm in diameter and ~10 nm in thickness. Study of the structure at the molecular level designates the composition and supports the nano-sized characteristics of the produced materials. The associated thermal properties combined with the corresponding optical properties suggest that the material may be used as a flame retardant filler with enhanced transparency. In this concept, the flame retardancy of composite coatings containing the produced nano-sized Mg(OH)_2_ was examined in terms of limiting oxygen index (LOI), i.e., the minimum concentration of oxygen that just supports flaming combustion.

## 1. Introduction

Flame retardant additives in coatings and bulk thermoplastic materials have been a subject of extensive research in the past [1]. Recent trends for the development of composite materials with improved thermal properties involve properly designed nano-structured fillers [2]. These new nano-enabled materials are not only superior to their micro-structured analogues in terms of their flame retardant behavior, but they may also exhibit additional important functionalities, such as the mechanical [3] and the optical [4]. Another crucial parameter for their utilization in commercial products (wood constructions, textiles, electric-electronic applications, aeronautic applications, etc.) is their overall production cost.

Metal hydroxides are frequently used to boost flame retardancy by being incorporated as fillers into several polymer systems [5,6,7]. Magnesium hydroxide (MH) belongs to this family of materials due to (a) the absence of hazardous byproducts emitted upon its combustion, as well as (b) its suitable processing temperatures (higher than those of other hydroxide counterparts, e.g., alumina trihydrate). Several relevant applications of MH have been reported as an additive in: electrical cables [8], building and restoration materials [9], etc. It has been reported that superfine MH nano-structures possess additional improved thermal properties. This is a consequence of the dependence of material physicochemical properties on the shape and the agglomerated state as well as on the size and the distribution of the related nano-particles [10]. Hernist et al. have shown that these features may be controlled by following appropriate preparation procedures [11]. Superfine MH powders can be obtained via various methods; the most common in industry are the hydration of magnesium oxide (MgO) [12] and the precipitation of a magnesium salt with an alkaline solution [11]. More elaborate processes such as through solvothermal modification [13], microwave-assisted synthesis [14], and electrolysis of an aqueous magnesium salt solution [15] have also been reported. Nevertheless, the latter methods may require long synthesis times and complicated equipment and/or processing conditions, which pose difficulties on scaling up the manufacturing of flame-retardant materials both from the technological and the economical viewpoint. 

Here, we propose a simple and a time effective methodology that enables the production of nano-structured flame retardant MH inorganic filler, nano-MH, with distinct thermal as well as optical properties from cheap mineral precursors, i.e., huntite and hydromagnesite. We present the required steps concerning the appropriate thermodynamic processes, and we provide a detailed characterization of all materials involved in the intermediate stages as well as the final product within the framework of a procedure at the laboratory scale. Formulations based on the nano-sized Mg(OH)_2_ and commercially available polymers and dispersants lead to the formation of coatings on prototype wooden samples, and the extraction of LOI enabled the characterization of the associated flame retardant properties. The methodology proposed has a scientific impact since it describes a way to produce MH hexagonal nano-platelets without following the typical hydrothermal pathways; in addition, it offers the potential for applications since it may be upscaled at the industrial level. 

## 2. Materials and Methods

Huntite mineral (coded H5) was kindly provided by Sibelco, while hydromagnesite mineral (coded LH15) was provided by LKAB GmbH. These nano-flame-retardant precursors are carbonate minerals, of hydro-magnesite [Mg_5_(CO_3_)_4_(OH)_2_·4H_2_O] and huntite [Mg_3_Ca(CO_3_)_4_]. They are both typically occurring as platy crystals. They exhibit relatively low refractive indexes (1.53 & 1.62, respectively), and most importantly, comparable to those of most polymer formulations where they will be incorporated, avoiding in that way to diversify any relevant transparency. They are both commercially available in the form of powder with particle size 3–10μm.

Lab-scale thermal treatment of the minerals (at 300, 400, and 500 °C) was performed for selected time intervals at an ambient atmosphere in an oven (Figure 1a). The thermal treatment step of the process (step A) was followed by hydration of the calcined powders (step B), which involved stirring their aqueous suspensions (4 g in 40 ml distilled water) at 90 °C for 4 h (Figure 1b). The hydrated material was collected after being dried in an oven at 90 °C for 20 h.

Thermogravimetric analysis (TGA) was performed by a TGA Q50 (TA Inst) instrument under inert conditions and at a 10 °C/min rate. The morphology of the materials was studied by a scanning electron microscope (SEM) Zeiss SUPRA 35 VP-FEG equipped with EDS (Bruker GmbH, Quanta200) and BSE detectors (K E Developments, Ltd., Cambridge, U.K.), operating at 5–20 keV. X-ray diffraction (XRD) measurements were performed in a Bruker AXS system (type: D8 Advance) equipped with a furnace. In situ and ex situ measurements revealed the components of the precursor materials as well as their structure. A Specac sampling accessory was used to measure Attenuated Total Reflectance (ATR) spectra. The spectra were recorded by using a Bruker Equinox FTIR system, with spectral resolution of 4 cm^−1^. Each spectrum resulted from the averaging of over 30 scans. For the recording of the Raman spectra, two types of instruments were used. Firstly, a Bruker FT-Raman instrument (FRA-106/S component attached to an EQUINOX 55 spectrometer), described in detail before [16]. Secondly, an HR-800 JY UV–Vis-Raman system (Horiba Scientific, Jobin Yvon, Villeneuve d′Ascq, France), where the excitation was performed by an air cooled He-Cd laser operating at 441.6 nm. The backscattered configuration was selected using a 50× (NA = 0.55) microscope objective. The spectral resolution was better than ~2.5 cm^−1^ for all measurements. The signal was recorded by a LN 2-cooled 2D-CCD detector. 

UV-Vis spectra were recorded from aqueous suspensions using a double beam UV/Vis Spectrometer of Hitachi (U-3000), while for the Dynamic Light Scattering (DLS) experiments a Malvern z-sizer was utilized. For the Differential Scanning Calorimetry (DSC) measurements the Q100 unit (TA Instruments, New Castle, DE, USA) equipped with a controlled liquid N_2_ cooling system was used. The heating rate selected was 10 °C min^−1^. Nitrogen physisorption was conducted at 77 K after sample degassing for 1 h at 200 °C (Autosorb-1, Quantachrome Instruments). The specific surface area was calculated using the standard BET method.

## 3. Results and Discussion

Both minerals consist of flakes in the sub-mm scale. The flakes’ lateral dimensions are typically in the range of ~2–10 μm; however, smaller particles (<2 μm) do exist with thickness close to the nano-meter range (Figure 2). The minerals are actually a natural mixture basically of hydromagnesite and huntite with LH15 composed mostly of hydromagnesite and H5 consisting primarily of huntite. Minor contribution of other minerals, e.g., CaCO_3_, CaMg(CO_3_)_2_, adds to the overall mass of the H5 and the LH15 samples.

Both hydromagnesite and huntite decompose through respective endothermic reactions. Hence, it is well known that the thermal decomposition of hydromagnesite starts at ~200 °C up to ~400 °C. In this temperature range, water molecules and hydroxyls existing in the structure are removed. The residual solid products are actually magnesium and calcium oxide and possibly magnesite in the case of a relatively low temperature decomposition process of hydromagnesite. The chemical reactions taking place are:Mg_5_(CO_3_)_4_(OH)_2_ 4H_2_O → Mg_5_(CO_3_)_4_(OH)_2_ + 4H_2_O(1)
Mg_5_(CO_3_)_4_(OH)_2_ → 2MgCO_3_ + 3MgO + 2CO_2_ + H_2_O(2)

Thermal treatment at temperatures > 400 °C leads to the decomposition even of the magnesite producing magnesium oxide:MgCO_3_ → MgO + CO_2_(3)

The thermal decomposition of huntite follows a different route:Mg_3_Ca(CO_3_)_4_ → 3MgO + CaO + 4CO_2_(4)
and takes place at the temperature interval 450–650 °C. Thus, hydromagnesite’s decomposition leads to the formation of MgO, while huntite’s decomposition ends at two different oxides, MgO and CaO, with 3/1 mol ratio.

### 3.1. Starting Materials

According to Hollingbery and Hull [17], the thermal decomposition of hydromagnesite is composed of several mechanisms depending on the test conditions. The minerals under investigation are hydromagnesite/huntite mixtures together with other crystal structures in minor proportions. Under an inert atmosphere, the TGA results (Figure 3) indicate only two decomposition stages for hydromagnesite attributed to the loss of the crystalline water (~200–300 °C), followed by the combined dehydroxylation and decarbonation (~350–480 °C). For huntite only, one peak is observed (~480–600 °C) associated with the loss of carbon dioxide from the carbonate groups associated with the magnesium ions. The decarbonation of calcite residues requires higher temperatures, which are not reached in the current study. However, magnesium calcite, magnesite, or even dolomite are being formed during the decarbonization process.

The thermal decomposition of the two types of precursor minerals is shown in the TGA graphs of Figure 3. The H5 mineral is mainly composed of huntite (steep loss of mass at ~566 °C attributed to dissociation of carbonate ions associated with Mg) with some contribution of hydromagnesite (smaller losses observed at ~200 °C and 400 °C). On the contrary, the LH15 mineral is dominated by hydromagnesite (major mass losses at 237 °C and 437 °C) with smaller quantities of huntite (minor mass losses at ~550 °C). For the hydromagnesite sites, dehydration is completed at temperatures <350 °C, while the decarbonation step takes place at temperatures <500 °C. The mass loss observed in the temperature range ~600 °C–700 °C is attributed to the dissociation of carbonate ions associated with Ca, possibly from dolomite impurities in the minerals. It is actually a lower decomposition temperature than those previously reported for natural dolomite under argon flow (which is typically within the 700–850 °C range) [18,19]. However, similar shifts of the observed step to lower temperatures have been observed by T.D. Humphries et al. [20] and were attributed to the heating rate, the quantity of impurities, and the dolomite crystal particle size.

Nearly 50–55% of the initial mass is released as carbon dioxide and water after heating to 700 °C. The mass loss derivative plots as a function of temperature are given in Figure 3b.

The FTIR spectra of the precursor minerals are shown in Figure 4a. In agreement with the literature, the normal modes of vibration assigned to hydromagnesite [21,22] and huntite [23,24] are resolved in the FTIR spectra of LH15 and H5 minerals (see also Table 1). The normal modes observed in the minerals’ spectra suggest that they are basically composed of different proportions of both huntite and hydromagnesite, which qualitatively agree with the TGA results.

Raman spectroscopy as well as X-Ray diffraction confirm the composition of the two minerals and indicate the presence of a third component, dolomite, in small proportions. The top spectrum of Figure 4b represents mineral LH15. The fitting of the CO_3_
^−2^ stretching spectral region, at ~1050–1130 cm^−1^, indicates the existence of at least three bands attributed to hydromagnesite/huntite, 1117 cm^−1^/1121 cm^−1^, and dolomite at ~1098 cm^−1^. A probable, very weak band at ~1087 cm^−1^ may be assigned to calcite. The XRD data confirm the existence of hydromagnesite in great proportions in mineral LH15, with huntite and dolomite only in smaller proportions (Figure 5).

The characterization of the H5 mineral revealed a composition similar to that of LH15. However, in the latter case, the primary structure was found to be huntite, with smaller contributions of hydromagnesite and dolomite. Since huntite and hydromagnesite are the two structures that undergo decomposition at the temperatures typically used by flame retardant materials, and taking into account that the temperatures reached for the production of the final nano-material are far below the decomposition of dolomite, we will consider its content in the starting materials as negligible. The relative concentration of hydromagnesite and huntite as extracted by XRD and TGA experiments for the H5 and LH15 minerals is given in Table 2.

### 3.2. Thermally Treated Minerals

The first process (step A) targeting the production of nano-MH describes the thermal decomposition of hydromagnesite and/or huntite minerals for the formation of nano-structured MgO. In the following sections, results referring to the LH15 precursor are principally given. The corresponding results for the alternate precursor, H5, are analogous. 

The understanding of the process associated with the decomposition is revealed by in situ temperature dependent XRD experiments. In this concept, data were collected from the starting materials at various temperatures up to 700 °C; a set of such data for the case of LH15 is given in Figure 6. XRD data verify that the raw material is composed primarily of hydromagnesite, followed by huntite, with a small contribution of dolomite (shoulder peak at ~31°). In accordance with the TGA results, the peaks attributed to hydromagnesite are suppressed at 300 °C and vanish at 400 °C, giving their place to a particularly broad peak at 42.6° and a much weaker peak at 36.7°, which are assigned to MgO. The intensity of this broad peak is rather low at 300 °C but progressively increases at higher temperatures. The width of the peak indicates the small, nano-size morphology of the corresponding MgO crystals. The latter is verified from the ex situ SEM images (Figure 7b–e), obtained after thermal treatment of LH15 for 1 h at 300 °C, 400 °C, and 500 °C. Nano-particles of MgO are clearly observed forming in some cases large agglomerates with spherical-like morphology (indicated by arrows in Figure 7). The dimensions of the nano-particles and the agglomerates exhibit a distribution ranging from few nm to approximately 200 nm. 

In situ temperature dependent XRD data reveal that the sharp peaks of huntite and dolomite crystals are dictating the diffractograms up to 600 °C. At 600 °C, the huntite peaks vanish and a weaker broad peak at 29.4 °C appears, attributed to calcite imperfect or nano-sized crystals. At this temperature, the dolomite peaks are the only sharp peaks resolved. At 700 °C, the diffractogram is characterized by (i) the absence of hydromagnesite, huntite and dolomite peaks, (ii) the intensity stabilization of the broad MgO peaks, (iii) the concurrent appearance of a small broad peak at ~31.5°, which together with the more intense band at ~37° are assigned to CaO. At 700 °C, the only existing peaks are those of MgO and CaO. Similar results were found by Raman spectroscopy applied on thermally treated LH15 samples at 500 °C, i.e., complete loss of hydromagnesite peaks, with the only weak bands recorded attributed to huntite and dolomite (Figure 4).

### 3.3. Hydrolysis of Oxides

Step B of the treatment process involves hydrolysis of the products obtained from step A. Hexagonal shaped nano-platelets dominate the SEM images of the hydrolyzed products, as shown in Figure 7f. The typical size of these nano-platelets is ~200 nm in diameter and ~10 nm in thickness, which are characteristic of Mg(OH)_2_ crystals in agreement with XRD and FTIR data.

The quality of the nano-sized material that results after step B is furthermore revealed by BET and dynamic light scattering experiments, the latter of which were applied on the respective aqueous suspensions. The DLS results obtained from the mineral precursors indicated average hydrodynamic diameter for the particles of ~400–500 nm. Corresponding measurements on the material obtained after the thermal treatment (500 °C) and the subsequent hydration indicated the average hydrodynamic diameter of the particles to be ~100 nm. A typical size distribution by number for the case of H5 mineral is shown in Figure 8a, while the corresponding nano-MH after calcination and hydration of the former is shown in Figure 8b. Analogous results were obtained for the LH15 mineral.

In addition, BET measurements (Table 3) gave a volume specific surface area (SSA) for the modified mineral of 120 m^2^/cm^3^, well above the limit of 60 needed to qualify a material as a nano-material, according to the EC recommendation for the NMs definition [25].

Comparative XRD data of the LH15 precursor, the product obtained after its thermal treatment at 500 °C, and the final hydrolyzed product are shown in Figure 5b. The raw mineral consists mainly of hydromagnesite and quantities of huntite and dolomite. The thermally treated mineral consists mainly of nano-structured MgO, with small quantities of huntite and dolomite and minor quantities of calcite, due probably to partial decarbonization of huntite at 500 °C. The hydrolyzed product contains predominantly Mg(OH)_2_ (brucite), with small quantities of huntite and dolomite as well as minor proportions of calcite and magnesite, the latter of which is probably a result of partial carbonization of MgO.

The XRD findings are also verified by FTIR spectroscopy. The hydromagnesite peaks (in the shadowed areas of Figure 9) are absent in the thermally treated sample. On the other hand, the huntite peaks persist, while in the hydrolyzed product the most characteristic peak is that of brucite, at ~3700 cm^−1^.

The quantity of nano-MH formed strongly depends on the calcination process, most critically on the temperature upon which it takes place. Experiments that involve different treatment temperatures during step A of the process (ranging for example from 300 °C to 500 °C) are characterized by fractional calcination of the minerals that increases with temperature, resulting in different amounts of MgO. The generation of the subsequent step B of the process hydrolyzes the magnesium oxide to different concentrations depending on the parameters followed in step A. This is demonstrated in Figure 10 where XRD data from samples calcined in step A at different temperatures, 300 °C, 400 °C, and 500 °C, and then hydrolyzed indicate systematic Mg(OH)_2_ intensity enhancement (peaks at 18.5° and 38°).

### 3.4. Thermal and Optical Properties of the Synthesized Nano-Mg(OH)_2_

DSC thermograms of the initial mineral LH15 and the material produced after step B (hydration) can be seen in Figure 11. The thermal decomposition of hydromagnesite in the LH15 mineral may be monitored by the two endothermic peaks at ~304 and 441 °C associated with the losses of water molecules, hydroxyls, and carbon dioxide in direct analogy with the results obtained from the TGA experiments (orange curves in Figure 3). The major component of the product collected after the application of process step B is nano-Mg(OH)_2_, a fact that explains the intense endothermic peak shown at 396 °C in the respective thermogram. The ~30 °C shift to lower temperatures compared to the endothermic peak of the brucite crystals is a consequence of the nano-crystals formed via the employed method. The DSC results of the produced material agree well with the corresponding TGA results (Figure 3).

The decomposition heat for the aforementioned samples is included in Table 4. Compared to the values for Mg(OH)_2_ crystal found in the literature (which span a wide range of values), the corresponding decomposition heat for the case of the sample received after step B lies at the lower range [26]. This may be attributed to the fact that the material received after step B of the process is not pure Mg(OH)_2_ (it contains, huntite ~19% and dolomite).

Comparison of the DSC measurements of the mineral precursor—LH15—and of the nano-material—nano-Mg(OH)_2_—produced after thermal treatment at 500 °C, with subsequent hydrolysis for 4 h at 90 °C, indicates a slight increase (6% increase) in the total integrated endothermic heat flow of the Mg(OH)_2_ relative to the precursor. Moreover, this endothermic peak is located at a narrower and eventually beneficial temperature range in relation to the wood ignition temperature (300–364 °C depending on the wood type) [27].

The optical properties of the material produced are supreme compared to the initial minerals (Figure 12a). Optical spectra enabled the direct comparison of light extinction measured through suspensions of 0.1 mg/ml (Figure 12b). The results are in favor of the produced nano-material since it is at least an order of magnitude more transparent than the initial mineral (20 times in the visible).

### 3.5. Flame Retardant Properties

The actual flame retardant properties of the synthesized nano-MH, which may favor applications involving, for example, coatings for wooden construction materials, require several more parameters to be set in advance. Perhaps the most critical is its dispersion in formulations involving polymers that will effectively form the desired coating. Experiments were undertaken in this direction after producing formulations consisting of three typical commercially available polymers: (a) one described as Ecrothan 4075 (waterborne polyurethane emulsion based on polyester-polyol), (b) Ecrovin LV 340 eco (terpolymer emulsion based on vinyl acetate/vinyl versatate/maleic acid ester) and (c) LV 372 ADN (terpolymer emulsion based on vinyl acetate/vinyl versatate/acrylic acid ester).

Due to the alkaline nature of nano-Mg(OH)_2_, the formulations exhibited a strong tendency to flocculate. Dispersants had to be used in order to minimize it such as the commercially available PANX (ECRONOVA). The application of the formulations on prototype wooden specimens and the subsequent tests provided evidence on the flame retardant properties of the coatings. The tests were based on standard procedures that estimate the Limited Oxygen Index (LOI), typically defined as the minimum concentration of O_2_ in a N_2_/O_2_ mixture that is required to just support combustion of the test sample for 3 min or to consume a length of 5 cm from the sample. In this concept, the higher the LOI value, the greater the fire retardancy. Equation (5) was used for the extraction of LOI:LOI = 100 ([O_2_]/([O_2_] + [N_2_]))(5)
where [O_2_] and [N_2_] were the concentrations of oxygen and nitrogen inserted in the testing compartment and selected in such a way as to meet the requirements of the definition of LOI.

Details related to the wooden samples prepared and tested along with the evaluated LOI are given in Table 5. Tests were accomplished for wooden samples coated with only the polymer dispersion, using formulations after adding the H5 or the LH15 minerals as received and after adding nano-Mg(OH)_2_ together with the dispersant. The LOI values shown in the table are average values of three individual tests.

The LOI results are those utilized for the screening of the flame retardant properties of the materials in the lab scale. When only the polymer formulation was applied on pieces of wood, the obtained results were similar to the un-coated wood (LOI for un-coated wood is ~25%), while when a mixture of the polymer formulation and of the mineral was utilized, an increase of the LOI occurred, as it is shown in Table 5, where results with the polymeric formulation of Ecrovin LV 340 eco are shown. The loading of the various coatings is in the same range but not identical; more specifically, the loading of the MH-containing coating is 6–12% lower than the H5 and LH15 coatings, and this could be responsible for the slightly smaller value of LOI in the former case.

## 4. Conclusions

We propose a process to produce nano-sized Mg(OH)_2_ from minerals of huntite and/or hydromagnesite. The final product possesses good thermal properties within the temperature window that is considered to be of interest for flame retardant materials used on wood. In addition, its optical properties are superior with respect to the initial minerals, and they are in favor when transparency is considered important. Formulations based on the nano-sized Mg(OH)_2_ produced after application of the above mentioned process were prepared using commercially available polymers, and they were applied on wooden samples. The use of a dispersant was essential for the formation of coatings of good quality. All tests indicated adequate flame retardant properties. Optimization of the final formulations is required, and it may result in coatings with improved flame retardant properties.

## Figures and Tables

**Figure 1 nanomaterials-12-02433-f001:**
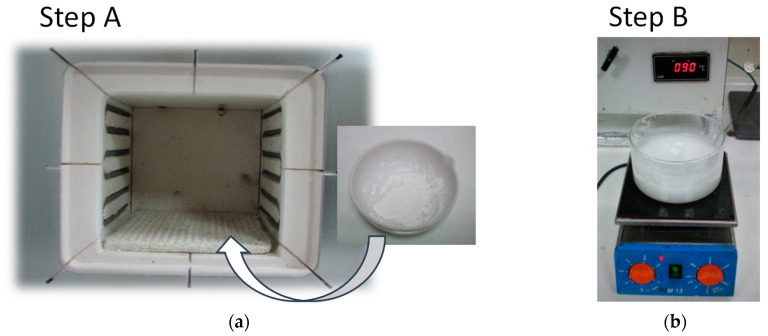
(**a**) Chamber furnace used for the thermal treatment of the minerals under ambient atmosphere at selected temperatures (step A). (**b**) Hydration process of the materials under stirring and controlled temperature (step B).

**Figure 2 nanomaterials-12-02433-f002:**
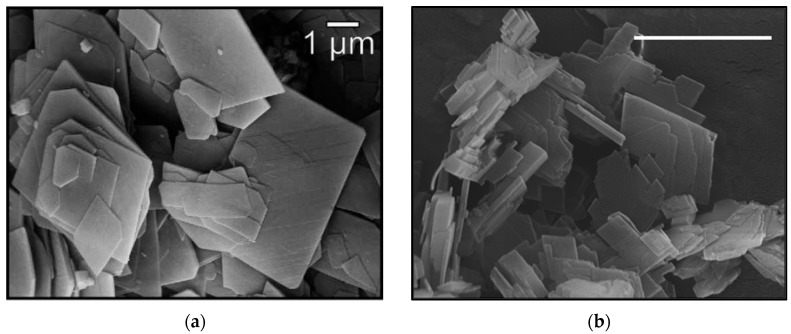
SEM images of (**a**) LH15 and (**b**) H5 minerals (scale bars indicate 1 μm).

**Figure 3 nanomaterials-12-02433-f003:**
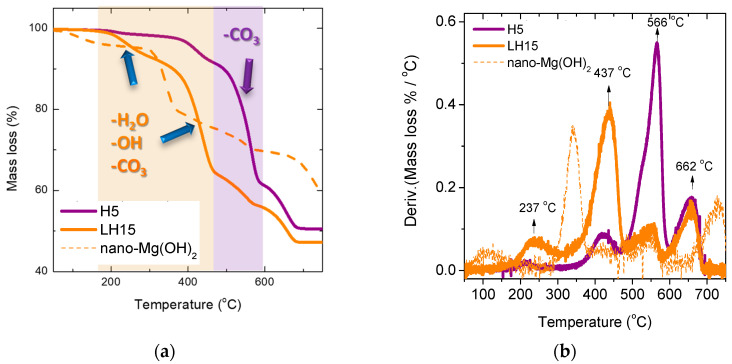
(**a**) TGA measurements for the two minerals of different composition: H5 mostly huntite and LH15 mostly hydromagnesite. Highlighted regions indicate the temperature range reached through the thermal decomposition processes followed in our experiments. (**b**) The mass loss derivative as a function of temperature. Dotted curves are obtained from the material collected after step B (i.e., mainly nano-Mg(OH)_2_).

**Figure 4 nanomaterials-12-02433-f004:**
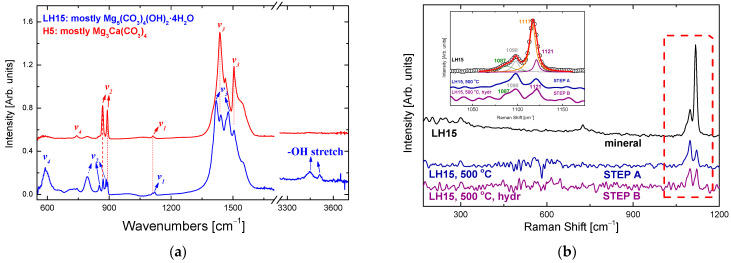
(**a**) FTIR spectra of the precursor minerals. The dominant hydromagnesite and huntite bands are annotated. (**b**) Raman spectra of the LH15 precursor mineral as well as the resultant materials after step A (thermal treatment at 500 °C) and step B (hydration at 90 °C). A fitting in the spectral region of the CO_3_^2−^ vibrational modes is given as an inset. Assignment of the four bands is given in the text and Table 1.

**Figure 5 nanomaterials-12-02433-f005:**
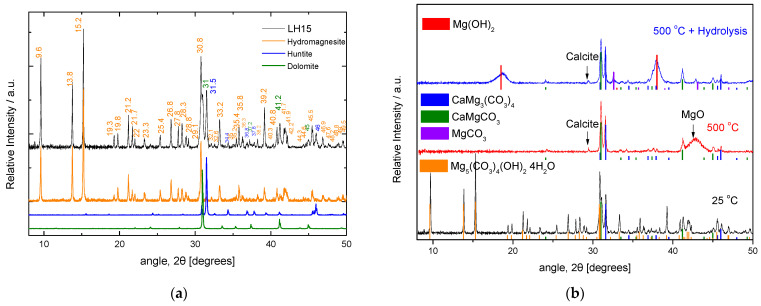
(**a**) The LH15 X-Ray diffractogram (on top) along with three colored diffractograms at the bottom, which correspond to the XRD mineralogical library data for hydromagnesite, huntite, and dolomite. (**b**) XRD diffractograms of the LH15 precursor material (at the bottom) and the material obtained after thermal treatment at 500 °C (in the middle) and after thermal treatment and subsequent hydrolysis (on the top). Colored lines indicate the position of the peaks for the corresponding crystal structures denoted on the left-hand side of the plot.

**Figure 6 nanomaterials-12-02433-f006:**
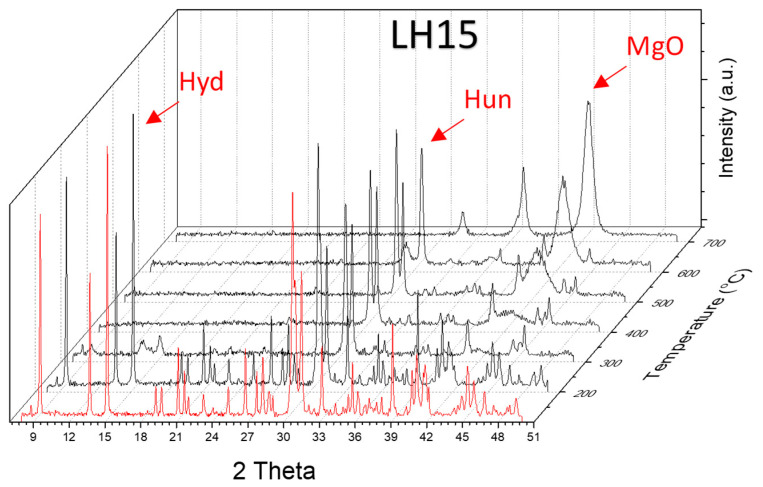
In situ temperature dependent XRD data obtained from LH15.

**Figure 7 nanomaterials-12-02433-f007:**
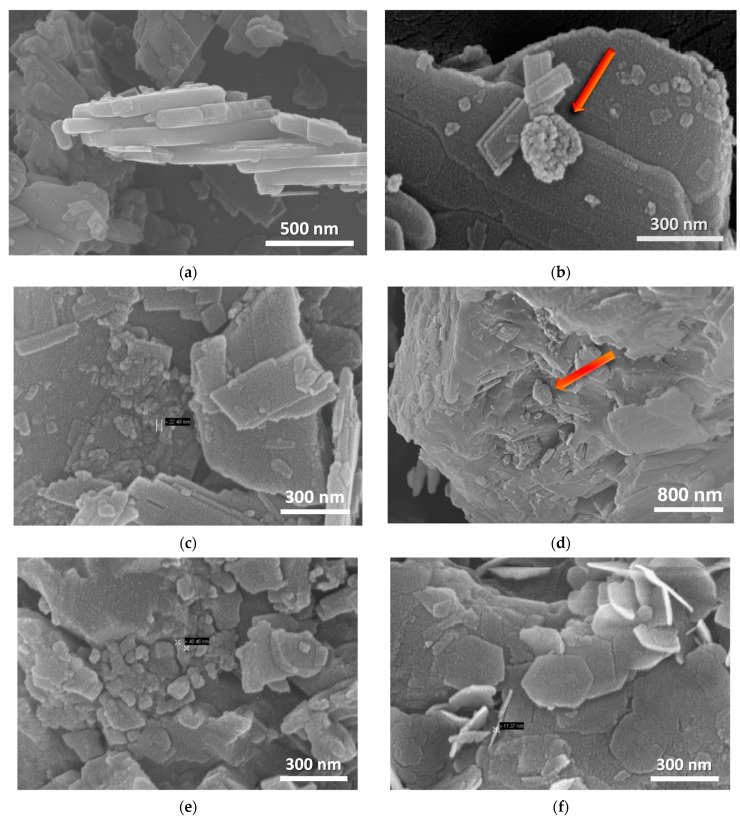
SEM images of LH15 (**a**) raw material, (**b**) after thermal treatment at 300 °C for 1 h, (**c**,**d**) at 400 °C for 1 h and (**e**) at 500 °C. for 1 h. (**f**) Sample shown at (**e**) after its hydrolysis.

**Figure 8 nanomaterials-12-02433-f008:**
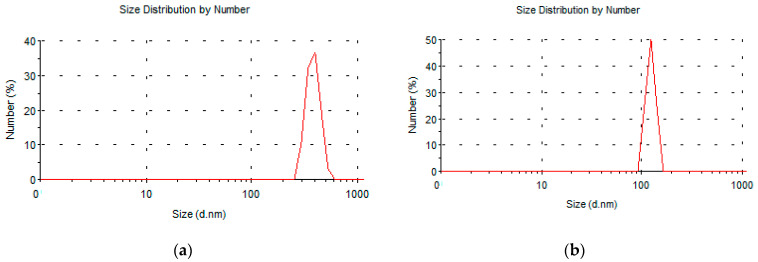
Size distribution by number of particles in an aqueous solution of (**a**) the H5 mineral and (**b**) the Mg(OH)_2_.

**Figure 9 nanomaterials-12-02433-f009:**
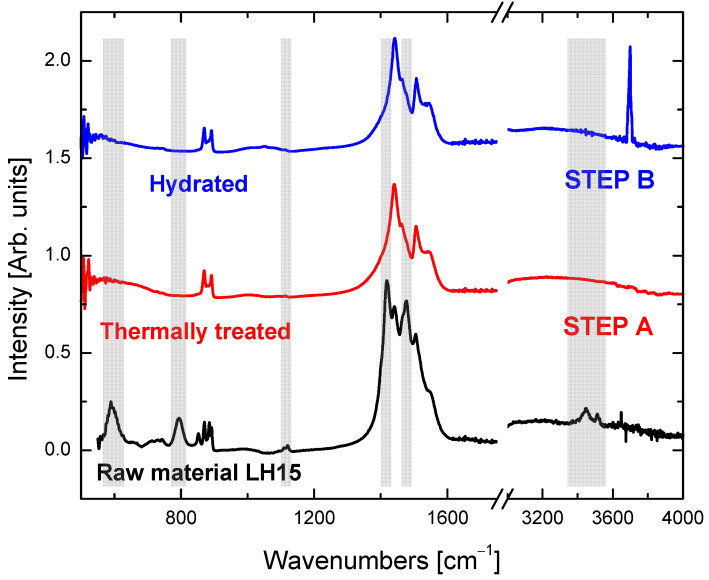
FTIR/ATR spectra of the samples whose XRD data are shown in Figure 5b. Shadowed regions indicate the vibrational modes of hydromagnesite.

**Figure 10 nanomaterials-12-02433-f010:**
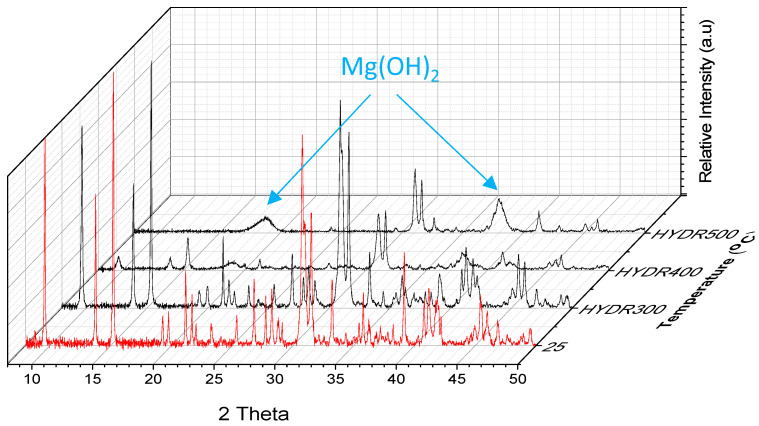
XRD data obtained at ambient conditions from the pristine mineral LH15 and after its treatment at 300 °C, 400 °C, and 500 °C (step A), with a subsequent hydrolysis (step B).

**Figure 11 nanomaterials-12-02433-f011:**
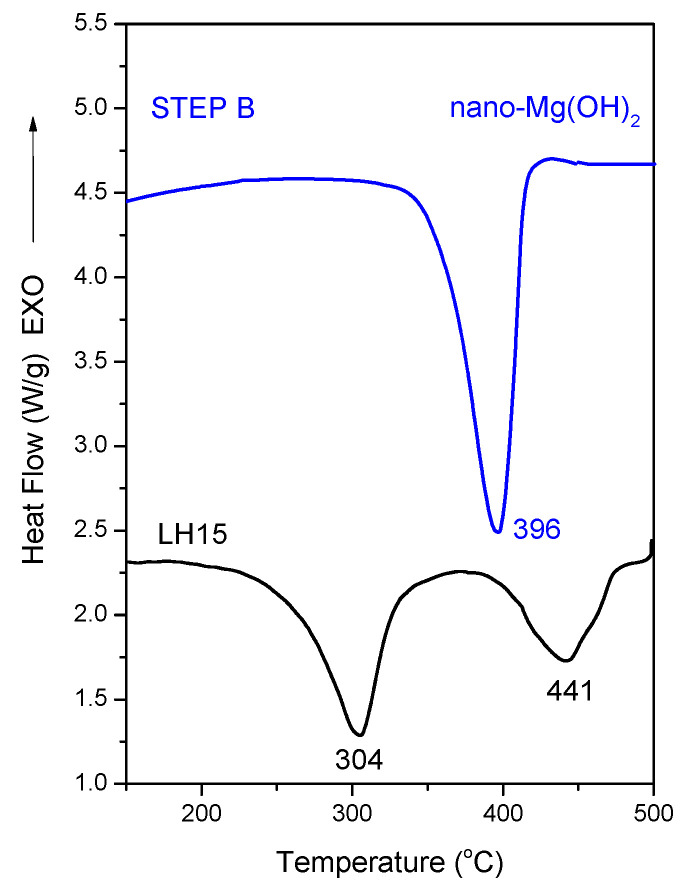
DSC thermograms of the LH15 mineral and the produced nano-Mg(OH)_2_ based material. The curves are shifted for clarity.

**Figure 12 nanomaterials-12-02433-f012:**
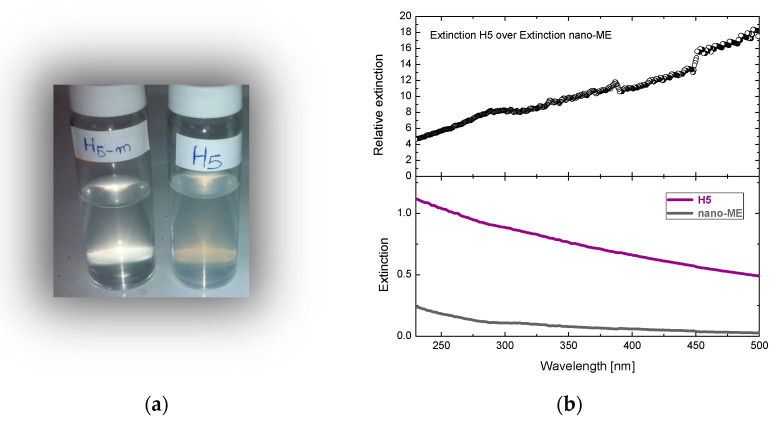
(**a**) Suspensions (0.1 mg/ml) of nano-Mg(OH)_2_ and the initial mineral H5. (**b**) Extinction spectra of the corresponding suspensions (**bottom**) and the relative extinction between them (**top**).

**Table 1 nanomaterials-12-02433-t001:** Assignment of the carbonate ion vibrational modes observed in the IR and Raman spectra of the H5 and LH15 minerals and their derivatives. The OH stretching vibrations assigned to hydromagnesite and brucite are also given.

Normal Mode	IR Bands [cm^−1^]			Raman Bands [cm^−1^]			
	Huntite	Hydromagnesite	Brucite	Huntite	Hydromagnesite	Dolomite	Calcite
*ν* _3_	1438	1419					
	1506	1477					
*ν* _1_	1112	1120		1117	1121	1098	1087
*ν* _2_	868	795					
	891	853					
		885					
*ν* _4_	743	594					
*-OH stretching*		3447	3698				
		3512					

**Table 2 nanomaterials-12-02433-t002:** Relative concentration of Huntite–Hydromagnesite in the precursor minerals.

Mineral	Huntite [%]	Hydromagnesite [%]
H5	88	12
LH15	19	81

**Table 3 nanomaterials-12-02433-t003:** Volume-Specific Surface Area for H5 and nano-Mg(OH)_2_.

Filler	SSA (m^2^/g)	VSSA (m^2^/cm^3^)
H5	18	49
Nano-Mg(OH)_2_	60.3	120

**Table 4 nanomaterials-12-02433-t004:** Decomposition heat of LH15 and nano-Mg(OH)_2_. The characteristic temperature of each peak is given in brackets.

	Decomposition Heat [J/g]		
Minerals	H_2_O Removal	OH Removal	Total
LH15	288.1 [304 °C]	163.2 [441 °C]	451.3
Hydr500		476.1 [396 °C]	476.1

**Table 5 nanomaterials-12-02433-t005:** The LOI values of wooden samples coated with Ecrovin LV 340 eco coating and its formulations with H5 and LH15 minerals as well as the produced nano-MH.

Formulation Description	Wood Sample Weight (g)	Total Sample Weight (g)	Loading g/m^2^	LOI (%)
LV 340 ECO	3.19	4.44	462	24
LV 340 ECO + H5 (50/50%wt)	3.60	5.13	568	34
LV 340 ECO + LH15 (50/50%wt)	3.31	4.73	526	31
LV 340 ECO + nano-MH + PANX (49.8/49.8/0.4%wt)	3.50	5.06	494	30

## Data Availability

Not applicable.

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
