# Peer review of "Flame Retardant Nano-Structured Fillers from Huntite/Hydromagnesite Minerals"

_nanomaterials, 2022, doi:10.3390/nano12142433_

Round 1
Reviewer 1 Report
In this paper, the author investigates new process to synthesize nano-structured Mg(OH)2. It is understandable that it is industrially beneficial, but there are doubts about its novelty. Therefore, I labeled this revision as major. Detailed comments are provided below:
i) I understand the industrial benefits, but what is the academic novelty?
ii) In Figure7, the agglomerated nano-particles are clearly observed by SEM. Is there agglomerate in the sample after thermal treatment of LH15 for 1h at 500 C? What about other processed? In addition, do agglomerated nano-particles spread to all of the samples?
iii) In Talbe5, commas and periods are mixed. Also, please check the significant figures.
iv) LOI of LV 340 Eco + H5 is higher than that of LV340 ECO + LH15 and LV340 ECO + nano-ME + PANX. Why is that? In Figure 3, the temperature of the peak of mass loss is different. Is this not related to LOI?
Reviewer 2 Report
This paper has the aim to produce nano-sized Mg(OH)2 from minerals of huntite and/or hydromagnesite. The final product possesses good thermal properties within the temperature window which is considered to be of interest for flame retardant materials used on wood. All tests indicated adequate flame retardant properties.
The subject is highly interesting and such a paper should be taken into account.
The authors used performant investigation techniques, very sensitive in order to put into evidence the synthesis and the applications.
The paper is well organized, and well documented so, in conclusion, I consider that this paper could be published in the present version.
However, there are some minor corrections that should be done. The figures are not very clear (for example Fig 4a, Fig.12a) and they should be redrawn.
Author Response
Reviewer 2.
Comments and Suggestions for Authors
This paper has the aim to produce nano-sized Mg(OH)2 from minerals of huntite and/or hydromagnesite. The final product possesses good thermal properties within the temperature window which is considered to be of interest for flame retardant materials used on wood. All tests indicated adequate flame retardant properties.
The subject is highly interesting and such a paper should be taken into account.
The authors used performant investigation techniques, very sensitive in order to put into evidence the synthesis and the applications.
The paper is well organized, and well documented so, in conclusion, I consider that this paper could be published in the present version.
Comment
However, there are some minor corrections that should be done. The figures are not very clear (for example Fig 4a, Fig.12a) and they should be redrawn.
Our Answer
We thank the reviewer for his/her positive remarks concerning our work. Following his/her suggestion we have made appropriate alterations in Figures 4a, Fig. 8 and Fig. 12a and hope that the quality of the revised figures is satisfactory.
Round 2
Reviewer 1 Report
The author have properly answered and corrected the reviewer's questions, so I labeled this reision as accept.